# Exposure to Quaternary Ammonium Compounds Selects Resistance to Ciprofloxacin in *Listeria monocytogenes*

**DOI:** 10.3390/pathogens10020220

**Published:** 2021-02-18

**Authors:** Alizée Guérin, Arnaud Bridier, Patricia Le Grandois, Yann Sévellec, Federica Palma, Benjamin Félix, Sophie Roussel, Christophe Soumet

**Affiliations:** 1Fougères Laboratory, Antibiotics, Biocides, Residues and Resistance Unit, French Agency for Food, Environmental and Occupational Health (ANSES), 35133 Fougères, France; alizee.guerin@anses.fr (A.G.); patricia.legrandois@anses.fr (P.L.G.); christophe.soumet@anses.fr (C.S.); 2Maisons-Alfort Laboratory of Food Safety, University Paris-Est, French Agency for Food, Environmental and Occupational Health (ANSES), 94700 Maisons-Alfort, France; yann.sevellec@anses.fr (Y.S.); federica.palma@anses.fr (F.P.); benjamin.felix@anses.fr (B.F.); sophie.roussel@anses.fr (S.R.)

**Keywords:** Listeria monocytogenes, biocides, antimicrobial resistance, bacterial adaptation, benzalkonium chloride, didecyl dimethylammonium chloride, ciprofloxacin

## Abstract

In this contribution, the antimicrobial susceptibility toward 11 antibiotics and four biocides of a panel of 205 Listeria monocytogenes (Lm) strains isolated from different ecological niches (i.e., food, animals and natural environment) was evaluated. The impact of exposure to biocides on the antibiotic susceptibilities of Lm was also investigated. Lm strains isolated from food exhibited overall a lower susceptibility (higher minimal inhibitory concentrations, MIC) for ammonium quaternary compounds (QACs) and peracetic acid (PAC) than strains isolated from animals and natural environments. Conversely, the ecological origins of Lm strains did not significantly affect their susceptibilities towards antibiotics. Interestingly, repeated exposure to QACs recurrently led to a decrease in susceptibility toward ciprofloxacin (CIP), a fluoroquinolone antibiotic, largely used in human medicine. Moreover, these lower levels of susceptibility to CIP remained stable in most Lm strains even after subcultures without biocide selection pressure, suggesting an adaptation involving modifications at the genetic level. Results underlined the ability of Lm to adapt to biocides, especially QACs, and the potential link between this adaptation and the selection of resistance toward critical antibiotics such as ciprofloxacin. These data support a potential role of the extensive use of QACs from “farm to fork” in the selection of biocide and antibiotic resistance in pathogenic bacteria such as Lm.

## 1. Introduction

Listeria monocytogenes (Lm) is an important cause of bacterial foodborne infections in Europe with a significant increase in the prevalence of listeriosis cases for a decade [1]. Listeriosis mainly affects pregnant women and the developing fetus, elderly people and immunosuppressed individuals [2]. The human transmission is mainly due to the consumption of contaminated food products such as ready-to-eat meals, fish, meat, unpasteurized dairy products, fruits and uncooked vegetables [3,4]. This is intimately related to the ability of some Lm strains to adapt and survive in various environments along the food chain from natural environments to food products. Indeed, Lm is a ubiquitous bacterium and is commonly found in natural environments especially in soil, which constitutes a central ecological node between the environment, animals and the food industry [5]. A broad understanding of the global ecology of Lm and especially of the strategies of adaptation to its various ecological niches will be critical for better control of this foodborne pathogen. The 3-year research project “adaptive traits of Listeria monocytogenes to its diverse ecological niches” (LISTADAPT) in the frame of H2020 “One Health” European Joint Programme aimed to address this issue by identifying genetic markers underlying the adaptation of Lm to various environments. For this purpose, a wide collection of various Lm strains isolated from natural environments, animals and the food chain over 20 years was constituted as previously described [6]. We focused here on the adaptation of Lm to biocides and the consequence of such adaptation on antibiotic resistance, which constitutes a pressing public health issue. Biocides are used daily in the food chain to limit contamination of food products and reach microbiological quality requirements found on hygiene food law (EC) no. 852/2004 [7]. Biocides are defined as active substances or preparations with a chemical or biological action with a high toxicity aiming to eliminate, repel and control the quantity of undesirable organisms [8]. Most common biocidal substances are quaternary ammonium compounds (QACs), alcohols, aldehydes, peracetic acids or chlorine compounds. They have a broad spectrum of antimicrobial action with multifactorial modes of action compared to antibiotics [9,10]. Most of these agents target different components and biosynthesis pathways of the bacterial cell such as cell wall synthesis, bacterial membrane, certain steps of protein synthesis, DNA and RNA [11]. Under real conditions of application (presence of biofilms, organic matter or misuse of biocide), the exposure to lower concentrations of biocides may contribute to the appearance of resistance of Lm to antibiotics and/or biocides [12]. QACs are widely use as disinfectants in healthcare settings and food processing plants [13], and several studies showed that they have the potential to select resistance toward various antibiotics. Impact of exposure to benzalkonium chloride (BC) on susceptibility to various antibiotics such as ciprofloxacin, gentamicin or kanamycin in Lm was indeed reported in previous works [14,15]. However, there was less information concerning potential impact of other QAC such as didecyl dimethylammonium chloride (DDAC) and other biocide classes on antimicrobial resistance profiles of Lm strains. A better understanding of the ability of Lm to adapt to biocides is, thus, of prime importance to understand the emergence of antibiotic resistant strains in this species. 

In this context, this work investigated the impact of the food industry’s most representative biocides on the generation of antimicrobial resistance in Lm. Antimicrobial susceptibility profiles to four biocides and 11 antibiotics were determined and compared in a set of 205 Lm strains of diverse sources in order to detect differences by ecological niches. Further, we evaluated whether exposure to biocides have the potential to modify the antibiotic and biocide susceptibility profiles of these strains and the stability of these modifications. 

## 2. Results

### 2.1. Biocides and Antibiotic Susceptibility Profiles of the 205 Lm Strains

Distributions of biocide minimal inhibitory concentrations (MICs) revealed overall a low variability among the 205 Lm strains (Figure 1). MIC ranged from 0.63 to 5 mg·L^−1^ for BC and from 0.63 to 2.5 mg·L^−1^ for DDAC with a majority of strains at 1.25 mg·L^−1^ (115 strains, 56%) and 0.63 mg·L^−1^ (159 strains, 78%), respectively. MICs above the highest tested concentrations 10 mg·L^−1^ for DDAC were obtained for one strain. MICs of sodium hypochlorite (SH) are between 625 and 1250 mg·L^−1^ for 204 among the 205 Lm strains. MIC ranged from 156.25 to 625 mg·L^−1^ for peracetic acid (PAC) with 125 strains (61%) at 312.5 mg·L^−1^. MIC values for both QACs and PAC in Lm strains isolated from food products were overall higher than MIC values obtained for Lm strains from animal or environment (Fisher’s exact test, *p* < 0.00001). No significant difference was observed among SH MIC values depending on ecological origins of Lm strains.

MIC value distributions varied depending on antibiotic, with an MIC amplitude ranged from four- to 16-fold among the 205 Lm strain panel (Table 1). Epidemiological resistance thresholds (ECOFF: epidemiological cut-off defined by EUCAST: European Committee on Antimicrobial Susceptibility Testing) were available for four antibiotics: ampicillin (AMP), erythromycin (ERY), meropenem (MER) and tetracycline (TET) and indicated by vertical line in Table 1. Such thresholds are usually used to define strain as wildtype or non-wildtype in terms of resistance, but we only used these values here to discuss MIC distribution, since MIC determination protocol slightly differs from the EUCAST standard protocol. There was no strain above the ECOFF for AMP and ERY. For one strain, MIC of MER (0.5 mg·L^−1^) was twofold higher than the ECOFF (0.25 mg·L^−1^), and 35 strains displayed MIC of TET higher than the 1 mg·L^−1^ ECOFF including 34 strains with MIC of 2 mg·L^−1^ and one strain with an MIC superior to 16 mg·L^−1^. Tentative ECOFF (TECOFF) are also available for gentamicin (GEN) (2 mg·L^−1^) and CIP (4 mg·L^−1^) and are represented by vertical dotted line in Table 1. All the 205 Lm strains displayed values below this TECOFF for GEN with a maximum MIC value of 1 mg·L^−1^, and 2 Lm strains displayed a CIP MIC value of 8 mg·L^−1^, one dilution higher than TECOFF. Note that the origin of strains did not affect the antibiotic susceptibility profiles among the Lm strains panel (data not shown).

### 2.2. Adaptation to Biocide after Repeated Exposures to BC, DDAC, SH and PAC

Among the panel of 205 strains, 28 Lm have been selected to evaluate the effects of repeated exposure to biocides on MIC values. These 28 strains with various antimicrobial profiles and Lm ScottA stain were chosen to be representative of the full panel in terms of ecological origins and relevant in terms of biocide and/or antibiotic susceptibility profiles (Appendix A). The distributions of MIC of BC, DDAC, SH and PAC for the 28 Lm strains after repeated exposure to these biocides (and controls) are presented using box-and-whisker plots in Figure 2. Results showed a significant evolution of MIC of BC and DDAC for Lm strains exposed to both QACs compared to controls (paired T-test, *p* < 0.01). No significant changes in MIC of SH and PAC were observed due to the exposure to the four biocides.

### 2.3. Effects of Adaptation to Biocides on Antibiotic MIC Values

MIC values distribution of the 11 antibiotics for the 28 selected Lm strains exposed or not (control) to the four biocides were displayed using box-and-whisker plots in Figure 3. For the majority of antibiotics tested (AMP, ERY, GEN, MERO, vancomycin (VAN), chloramphenicol (CHL), streptomycin (STR), TET and tiamulin (TIA)), there is no significant difference between MIC distributions comparing Lm strains exposed to the four biocides and the control strains. A slight but significant (*p* < 0.01) increase in trimetroprim/sulfamethoxazole (TRS) MICs was detected in the 28 Lm strains exposed to DDAC, with a twofold increase in MIC distribution median between control and DDAC-exposed strains. Interestingly, results revealed a most pronounced difference in the distribution of CIP MICs between Lm strains exposed to both QACs (BC and DDAC) and those from the control panel. The most marked evolution was observed in strains exposed to DDAC with a significant increase in the median of CIP MIC distribution from 2.1 mg·L^−1^ (controls) to 9.48 mg·L^−1^. In a lesser extent, BC exposure led to an increase in the CIP MIC distribution median from 2.1 to 5.48 mg·L^−1^. The tentative ECOFF value for this antibiotic is 4 mg·L^−1^ (Table 1).

Figure 4A presented the distribution of CIP MIC in control strains and in Lm strains with CIP MIC increase before (full boxes) and after stability subcultures (stripped boxes). Significant differences (*p* < 0.01) were found between CIP MIC distributions of strains exposed to biocides (BC or DDAC), both before and after stability experiments when compared to the control Lm panel. Conversely, despite a slight decrease in distribution median for both QACs, no significant difference was observed between CIP MIC distributions of Lm strains before and after stability subcultures. 

Figure 4B,C, respectively, displayed the MIC ratios obtained for each strain exhibiting an increase in CIP MIC by comparing MIC after BC and DDAC exposure (full bars) and after stability experiments (stripped bars) to CIP MIC in corresponding control strains. CIP MIC ratios ranged from 2 to 4 after BC exposure (Figure 4B). After de-adaptation step, CIP MIC remained stable for four strains (FR-DA-U-UN-429, FR-ME-U-UN-465, NO-DEE-F2-63 and NL-GOA-UN-2), doubled for two strains (FR-ME-U-UN-446, FR-ME-U-UN-414) and were divided twofold for the eight other strains (but still at least twofold superior to MIC of corresponding control strain). Concerning DDAC (Figure 4C), CIP MIC ratios compared to control panel mostly ranged from 4 to 8 after repeated exposure except strains FR-ME-U-UN-414 and FR-FI-U-UN-418 for which MIC doubled and strain ScottA A for which a 16-fold increase was observed. After the de-adaptation step, CIP MIC remained stable in 12 strains or even increased for three strains, up to four times for strain FR-ME-U-UN-414. CIP MIC were divided twofold for five strains (FR-ME-P-UN-410, FR-ME-U-UN-465, FR-FI-U-UN-445, SI-BOV-CP-I-143, NO-OTH-O-19) and fourfold for strains ScottA but still fourfold superior to MIC of corresponding control strains. These results underlined that the repeated exposure to BC or DDAC is able to promote stable changes in Lm strains susceptibility to CIP. 

## 3. Discussion

In this study, the susceptibility profiles to four biocides and 11 antibiotics of 205 Lm strains from different origins were firstly determined. Concerning biocides, the data collected are consistent with existing data from studies using comparable methodologies. Here, most Lm strains had an MIC value of between 0.63 and 5 mg·L^−1^ for BC. Yu et al. [15] showed similar BC MIC from 2 to 6 mg·L^−1^ for 25 Lm strains isolated from meat products and food production environments. Likewise, Mereghetti et al. [16] showed that among 97 Lm strains, BC MICs ranged from 1 to 4 mg·L^−1^ for 90 strains and were higher than 8 mg·L^−1^ in only seven strains. Concerning DDAC, in agreement with DDAC MIC values ranging from 0.63 to 2.5 mg·L^−1^ obtained in this study, we previously showed in 31 Lm strains isolated from pig feces and pork meat that MIC values varied from 0.5 to 1.5 mg·L^−1^ [17]. MIC of SH mostly ranged here from 625 and 1250 mg·L^−1^ slightly higher than the 500 mg·L^−1^ SH MIC reported by Bansal et al. [18] but below the 3500 mg·L^−1^ MIC reported by Rodriguez-Melcon et al. [19]. PAC MIC values ranged here from 156.5 to 625 mg·L^−1^ for the 205 Lm strains, consistently with MIC values previously reported from 115 to 2713 mg·L^−1^ [20]. Unlike antibiotics, there is no criteria (such as ECOFF) to distinguish between a susceptible and a resistant bacteria toward biocides. However, Morrissey et al. [21] evaluated MIC distributions for biocides including BC and HS against 3327 isolates belonging to pathogenic bacteria and proposed ECOFF for these biocides based on these distributions. Although they do not include Lm, they used some Gram-positive species such as Staphylococcus aureus, Enterococcus faecium and E. faecalis. Thereby, they defined ECOFFs for BC and SH at 16 and 4100 mg·L^−1^ for S. aureus and 8 and 8200 mg·L^−1^ for E. faecium and E. faecalis, respectively. MIC values obtained in this work are, thus, below these ECOFF values for both biocides, suggesting a higher susceptibility of Lm comparing to these species. However, it should be carefully interpreted, since slight differences in methodology can significantly affect MIC values, and no universal strains reference was used.

Interestingly, the comparison of MIC distributions for the four biocides revealed that Lm strains isolated from food mostly displayed lower susceptibility toward both QACs (BC and DDAC) and PAC compared to Lm strains isolated from animals or natural environments. These results suggest a connection between the frequent use of these biocides in food-processing industries, which results in a recurrent selective pressure and the modification of the susceptibility to biocides of bacterial populations exposed. In line with this, we observed that a high proportion of Lm strains displayed a decrease in their susceptibility to both QACs (BC and DDAC) after repeated exposure to these molecules. The number of in vitro studies indeed demonstrated the ability of bacteria including Lm to adapt to repeated biocide exposure, in particular to QACs [15,22]. In this study, the large panel of strains enabled us to confirm such potential in an unprecedented way by comparing a significant number of strains from various ecological niches. Together, these observations emphasize the importance of a careful use of biocides and the need to go further in our understanding of bacterial adaptation strategies.

Moreover, an increasing number of studies showed a link between adaptation to biocides and the decrease in susceptibility to antibiotics including Gram-positive species, but little is known about Lm [23]. A better understanding of this relation is, thus, required to prevent antibiotic resistance emergence and finally treatment failures [24]. In this study, we highlighted that repeated exposure to sublethal QACs (BC or DDAC) concentrations, as those encountered by bacteria in industrial settings for instance, can lead to an increase in CIP MIC values in Lm. Although methodology used slightly differed, it has been already observed that growing in presence of BC led to a decrease in susceptibility to CIP in Lm [14,15]. Interestingly, such an effect of BC exposure on CIP susceptibility was also recently demonstrated in E. coli [25], and we also showed that DDAC exposure could led to an increase in CIP MIC in E. coli [17]. However, it is the first time that such an effect of DDAC exposure on CIP MIC was demonstrated in Lm, with an eightfold MIC increase for several Lm strains. Moreover, this modification of CIP persisted even without biocide selection pressure for most strains, suggesting that exposure to QAC has influenced CIP resistance-related mechanisms at the genetic level.

These cross-adaptations often rely on the overexpression of nonspecific multidrug efflux pumps in Gram-positive bacteria [23]. Two well-described efflux pump systems play a role in the multiple resistances observed in Lm. The efflux pump MdrL may be involved in the export of antibiotics from the cell such as macrolides and cefotaxime as well as heavy metals and ethidium [26]. Several studies revealed the contribution of this efflux pump in QAC resistance (mostly BC) [27]. The second efflux pump is Lde, which was associated to fluoroquinolone resistance, acridine orange and ethidium [26]. The role of Lde efflux pumps system in resistance of CIP has already been shown in Lm [28,29]. Certain QAC-resistant strains of Lm seemed to be able to have an overexpression of efflux pumps as MdrL and Lde [14,26], and the role of efflux pump overexpression was also demonstrated in post-adaptational resistance to BC [30]. More recently, the efflux pump EmrE has been described. It was associated to the adaptation of Lm strains to BC, but this efflux pump did not appear to be related to a decrease in CIP susceptibility [31,32]. Here, the decrease in susceptibility to BC and ciprofloxacin observed several times in strains exposed to BC and DDAC may involve efflux pump systems as an unspecific response to biocidal exposure. In addition, it has been shown that some mutations in Lm strains exposed to QAC play a role in the modification of the bacterial cell permeability [30,33]. These types of mutations could also influence the susceptibility to other substances similar to antibiotics such as CIP. The genomic comparative analyses between QAC-adapted Lm strains and parental strains would help us to understand the nature of genomic modifications selected by QAC exposure and how such mutations affect susceptibility to CIP. 

To conclude, this work provides additional data on the susceptibility profiles of a large number of Lm strains from different environmental niches to different antibiotics and biocides. Moreover, it provides new evidences of the potential of biocides as QACs to select antibiotic resistance in bacterial pathogens such as Lm. In particular, we highlighted for the first time the relation between the exposure to DDAC sublethal concentrations and the selection in this species of resistance to CIP, a critically important antimicrobial according to WHO classification. These results underline the importance of an appropriate use of biocides, especially using adequate concentrations to preserve their efficacy and prevent the emergence of antibiotic cross-resistance.

## 4. Materials and Methods

### 4.1. Bacterial Strains and Growth Conditions

The study includes 205 strains of Lm selected from the wide European collection of Lm strains built in the framework of the “One Health European Joint Programme ListAdapt” (grant #773830; https://onehealthejp.eu/jrp-listadapt/ (accessed on 16 February 2021)) and previously described [6]. Lm strains in this study came from various ecological niches and were isolated in 16 different European countries: 102 strains from ready-to-eat food; 65 strains from different animal species; 38 strains from natural environment (strain information are listed in Appendix A). The Lm strains were grown at 37 °C on trypticase soy agar (TSA) plate or broth (TSB) with or without yeast extract (0.6%) and horse blood (5%). These strains were kept at −80 °C in 20% glycerol cryoprotective solution of conservation.

### 4.2. Minimal Inhibitory Concentrations (MIC) of Antibiotics and Biocides

The profiles of antibiotic resistance of the 205 strains were performed with the determination of MICs using a broth microdilution method. Eleven antibiotics were tested at various concentration ranges (mg·L^−1^) (abbreviations Eucast.org): ampicillin (AMP, 0.015–2); chloramphenicol (CHL, 1–32); ciprofloxacin (CIP, 0.25–32); erythromycin (ERY, 0.06–8); gentamicin (GEN, 0.03–4); meropenem (MER, 0.03−1); streptomycin (STR, 2–32); tetracycline (TET, 0.06–8); tiamulin (TIA, 8–64); trimetroprim/sulfamethoxazole (TRS, 0.008/0.15–1/19); vancomycin (VAN, 0.25–8). These MICs were determined in Sensititre™ custom microplate (FRA1ANS; Thermo scientific) using a method slightly adapted from EUCAST protocol. Briefly, all strains were cultivated on TSA with blood for 24 h at 37 °C. Two to three colonies were picked and resuspended in 5 mL of sterile water for all strains. These suspensions were adjusted to 0.5 Mac Farland (around 10^8^ CFU.mL^−1^). Then, 100 µL were added to 11 mL of broth (MHB) with blood. The distribution of 100 µL of inocula in all 96 wells of microplates was carried out with Sensititre AIM^TM^ distributor (ThermoFisher scientific), and these microplates were incubated at 35 °C for 24 h. The negative control was performed with TSB added to yeast extract and the positive control corresponding to the Lm ScottA strain. MIC corresponds to the lowest concentration resulting in the growth inhibition of bacteria in the well. 

Biocide resistance profiles of these strains were also performed through the determination of MICs by the broth microdilution method adapted from NF EN 1040. Lm ScottA was used as a reference for each experiments. Four biocides were used in this study at various concentration (mg·L^−1^): benzalkonium chloride (BAC) 50% consisted of 65–70% BAC-12 and 30–35% BAC-14 (BC, 0.02-20) (Stepan, N°CAS: 68391-01-5); didecyl dimethylammonium chloride 50% (DDAC, 0.01-10) (VWR, N°CAS: 7173-51-5); sodium hypochlorite 14% (SH, 9.8-10000) (VWR, N°CAS: 7681-52-9) and peracetic acid 40% (PAC, 4.9-5000) (VWR, N°CAS: 79-21-0). All strains were isolated in TSA with blood for 24 h at 37 °C. Two to three colonies were collected in 4 mL of TSB with yeast extract (0.6%) and were incubated for 24 h at 37 °C. Then, 300 µL of these cultures were inoculated in 10 mL of TSB with yeast extract incubated at 37 °C for 3–4 h. These bacterial suspensions were adjusted to 0.1 (+/− 0.02) optical density at 620 nm (OD620) with sterile physiologic peptone water. All wells of microplates (Dutscher, 96 wells, PS, F BOTTOM, CLEAR Greiner bio one) were filled with 20 µL of biocide range solutions 10X at various concentrations, and 160 µL of TSB was added to yeast extract with 20 µL of bacterial cultures at 0.1 OD620 diluted to 1/100, corresponding to the final concentration between 3 and 6.105 CFU.mL^−1^. The negative and positive controls were the same as those for antibiotic microplates. MIC was determined after incubating the microplate for 24 h at 37 °C. 

### 4.3. Adaptation Experiments to Four Biocides

Twenty-eight Lm strains with various antimicrobial profiles and Lm ScottA stain were chosen for the adaptation step. These strains were chosen to be representative of the full panel in terms of ecological origins and relevant in terms of biocide and/or antibiotic susceptibility profiles (Appendix A). These strains were exposed to four various biocides at sublethal concentrations determined accordingly to MIC experiments: BC (0.6 mg·L^−1^); DDAC (0.31 mg·L^−1^); SH (312.5 mg·L^−1^); PAC (156 mg·L^−1^ or 78 mg·L^−1^). For all strains, two or three bacterial colonies isolated in TSA with blood were used to inoculate 4 mL of TSB supplemented by yeast extract and incubated for 24 h at 37 °C. Then, 300 µL of these cultures were added to 10 mL of TSB with yeast extract and incubated at 37 °C for 4 h. One milliliter of biocide (10X) was added to 9 mL of TSB with yeast extract containing 100 µL of bacterial suspension diluted to 1/100 and incubated at 37 °C for 24 h. For each strain, one control was carried out with 100 µL of bacterial suspension diluted (1/100) and added to 10 mL of broth. This last step was repeated for 10 consecutive days with a renewal of the broth containing biocide solution. After that, adapted strains were conserved at −80 °C, and their MICs of antibiotics (11) and biocides (4) were performed.

After this adaptation step, a de-adaptation step was performed to assess the stability of potential changes in antimicrobial susceptibility in Lm strains. This step was similar to the adaptation step but without biocides in TSB supplemented by yeast extract for 10 days. The MICs were determined once again after this stabilization step for all Lm strains adapted to the four biocides.

## Figures and Tables

**Figure 1 pathogens-10-00220-f001:**
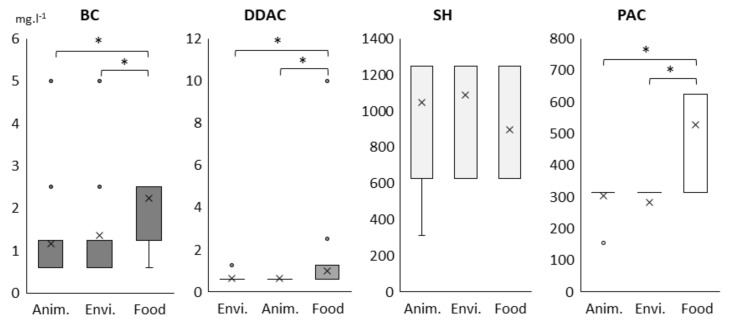
Distribution of minimal inhibitory concentration (MIC) values (mg·L^−1^) of the four biocides obtained for the 205 Lm strains depending on their ecological niches. Boxes range from the 25th to 75th percentile and whiskers extend below and above the box range from the lowest to the upper value, respectively. Cross indicates the median. Significant difference between distributions is indicated by star (Fisher’s exact test; * *p*-value < 0.00001). Cross (×) indicates the median, circles (∘) correspond to outlier values.

**Figure 2 pathogens-10-00220-f002:**
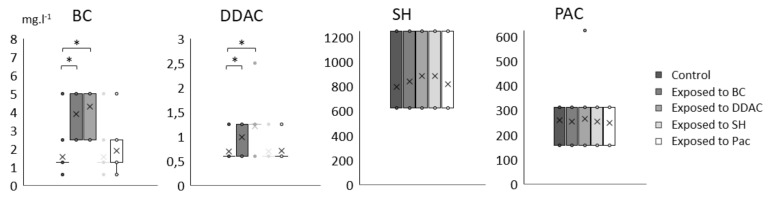
Variation of MIC values (mg·L^−1^) of benzalkonium chloride (BC), dimethylammonium chloride (DDAC), sodium hypochlorite (SH) and peracetic acid (PAC) for the 28 selected Lm strains exposed or not (control) to the 4 biocides. Significant difference with control is indicated by star (*, paired T-test, *p*-value < 0.01). Boxes range from the 25th to 75th percentile and whiskers extending below and above the box range from the lowest to the upper value, respectively. Cross (×) indicates the median, circles (∘) correspond to outlier values.

**Figure 3 pathogens-10-00220-f003:**
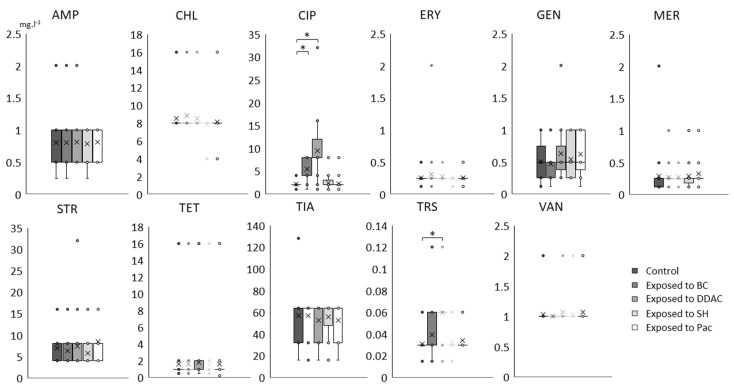
Variation of MIC values (mg·L^−1^) of the 11 antibiotics for the 28 selected Lm strains exposed or not (control) to the 4 biocides. Significant difference with control is indicated by star (*, paired T-test *p*-value < 0.01). Cross (×) indicates the median, circles (∘) correspond to outlier values.

**Figure 4 pathogens-10-00220-f004:**
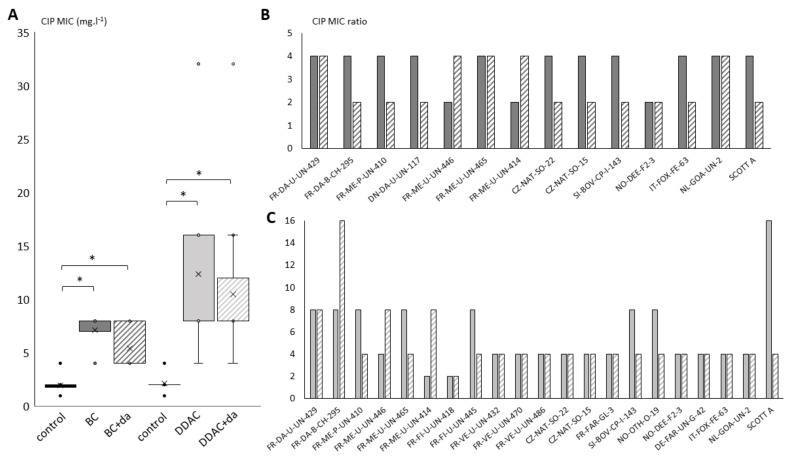
Stability of ciprofloxacin (CIP) MIC in Lm strains for which an increase in MIC was observed after exposition to BC (14 strains) and DDAC (21 strains). (**A**) Variation of CIP MIC values (mg·L^−1^) for Lm strains exposed to BC (dark grey) and DDAC (light grey). MIC values of corresponding strains unexposed to biocide for each biocide are also displayed (controls, black). Full bars correspond to values obtained after biocides exposition, and stripped bars correspond to values obtained after an additional de-adaptation step (+da), where strains were then subcultured in growth medium without biocides to assess the stability of the MIC increase. Significant difference with control is indicated by star (*, paired T-test *p*-value < 0.01). Cross (×) indicates the median, circles (∘) correspond to outlier values. (**B**) Increase MIC factor obtained for each Lm strain comparing CIP MIC after BC (B) or DDAC (**C**) exposure (full bars) and after de-adaptation steps (stripped bars) with CIP MIC of control Lm panel.

**Table 1 pathogens-10-00220-t001:** Distribution of MIC values (mg·L^−1^) of the 11 antibiotics obtained for the 205 Lm strains. The white zone corresponds to the ranges of concentrations tested, and the number of strains was indicated for each MIC for the different antibiotics. The vertical line (or vertical dotted line) corresponds to the ECOFF value (or tentative ECOFF) defined for Lm in Eucast.org when available.

	0.008	0.016	0.032	0.064	0.12	0.25	0.5	1	2	4	8	16	32	64	128
AMP						25	128	52							
CHL										1	192	12			
CIP							4	58	122	19	2				
ERY					22	178	5								
GEN				1	12	71	76	45							
MER					134	70	1								
STR								4		98	80	23			
TET						1	8	161	34			1			
TIA										1		5	53	144	2
TRS		20	176	9											
VAN								199	6						

## Data Availability

The data presented in this study are available on request from the corresponding author.

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
