# Peer review of "Exposure to Quaternary Ammonium Compounds Selects Resistance to Ciprofloxacin in Listeria monocytogenes"

_pathogens, 2021, doi:10.3390/pathogens10020220_

Round 1
Reviewer 1 Report
Alizee et al. showed the susceptibility to antimicrobials and biocides in Lm derived from various origins. However, the test methods of antimicrobials and biocides have non-ignorable problems (not used the recommended quality control strain, media and incubate condition).
Moreover, authors should show the genetic relevance about the ciprofloxacin after selection of biocide selection. Need more works.
Antimicrobial susceptibility tests
・In CLSI and EUCAT recommendation
media containing horse blood. (authors use sheep)
Incubate time: 24h (CLSI;20-24h , EUCAST;18±2h) (authors incubated the 24h, however breakpoints was used EUCAST)
Figure 2 was not figure, and authors should change the table.
Figure is difficult to understand.
Author Response
Alizee et al. showed the susceptibility to antimicrobials and biocides in Lm derived from various origins. However, the test methods of antimicrobials and biocides have non-ignorable problems (not used the recommended quality control strain, media and incubate condition).
Thanks for your comments. Although some tests did not stricly respect EUCAST recommendations (for instance 24h incubation with antibiotics instead of 18+/-2h as you mentionned), the experimental parameters were optimized for the reading of results in microtiter plate with the large panel of Listeria strains in our laboratory conditions. We assumed such modifications did not alter extensively MIC values based on our experience with Listeria in these conditons. Moreover, the goal of the study was the comparison between strains and all experiments were obviously strictly performed in identical conditions.
Moreover, authors should show the genetic relevance about the ciprofloxacin after selection of biocide selection. Need more works.
We agree that this point is actually the main perspective of the present work and will be the subject of a dedicated work currently beginning.
Antimicrobial susceptibility tests
・In CLSI and EUCAT recommendation
media containing horse blood. (authors use sheep)
Incubate time: 24h (CLSI;20-24h , EUCAST;18±2h) (authors incubated the 24h, however breakpoints was used EUCAST)
As mentionned previously, we did not strictly applied EUCAST recommendations to facilitate experiment interpretation/reading for the whole Listeria panel in our laboratory conditions and because that do not impact the comparison between the Listeria strains. We extended slightly the time of incubation from 18h to 24h because that facilitate the reading of the MIC results in microtiter plates and we used ScottA strain as reference because we already collected results with this strains in our lab conditions.
We apologize because we actually used horse blood in medium whereas we mentionned sheep blood in the manuscript. This was thus corrected in the revised manuscript (line 281).
Figure 2 was not figure, and authors should change the table.
Figure is difficult to understand.
Figure 2 was replaced by Table 1 in revised manuscript.
Reviewer 2 Report
In the submitted manuscript, Guérin et. al. investigated the impact of biocides exposure to the antibiotic susceptibility of Listeria monocytogenes. The authors first compared the minimal inhibitory concentrations of antibiotics and biocides (in particular, ammonium quaternary compounds and peracetic acid) among 205 strains isolated from three different ecological niches. The authors found food isolates have relatively higher biocides resistance than isolates from other two sources. Interestingly, the authors found that repeated exposure to QACs led to a decrease of susceptibility toward fluoroquinolone antibiotic such as ciprofloxacin. This study addresses the link between biocides exposure and antibiotic susceptibility of L. monocytogenes, which will provide valuable information to readers in the field who are interested in this topic. Overall the study is well conducted and manuscript is well written. I only have some minor suggestions for the authors to consider.
(1) line 97, please specify SH as “Sodium Hypochlorite”, since it is the first time SH shows in the manuscript.
(2) line 138: how these 28 strain were selected? The authors mentioned the selection process a little bit in the method. It will be helpful to address this briefly here in main text.
Author Response
In the submitted manuscript, Guérin et. al. investigated the impact of biocides exposure to the antibiotic susceptibility of Listeria monocytogenes. The authors first compared the minimal inhibitory concentrations of antibiotics and biocides (in particular, ammonium quaternary compounds and peracetic acid) among 205 strains isolated from three different ecological niches. The authors found food isolates have relatively higher biocides resistance than isolates from other two sources. Interestingly, the authors found that repeated exposure to QACs led to a decrease of susceptibility toward fluoroquinolone antibiotic such as ciprofloxacin. This study addresses the link between biocides exposure and antibiotic susceptibility of L. monocytogenes, which will provide valuable information to readers in the field who are interested in this topic. Overall the study is well conducted and manuscript is well written. I only have some minor suggestions for the authors to consider.
Many thanks for your carreful review and comments.
(1) line 97, please specify SH as “Sodium Hypochlorite”, since it is the first time SH shows in the manuscript.
The manuscrit was revised accordingly.
(2) line 138: how these 28 strain were selected? The authors mentioned the selection process a little bit in the method. It will be helpful to address this briefly here in main text.
As suggested, a sentence was added to clarify this point as suggested in the revised manuscript line 126.
Reviewer 3 Report
The authors survey a sizeable collection of L. monocytogenes from diverse sources and assayed their susceptibiltiy to antibiotics and biocides with the intent of identifying source specific trends in susceptibilities to these compounds of public health relevance. Overall the study is well conceceived and executed and well presented in their manuscript, I have only a few minor conceptual points and possible text corrections detailed below. My first relates to the attributes of the strain pool, while they were assembled from different sources what is known of their genotypes as this can have significant impacts on functional attributes of L. monocytogenes. Not knowing even the serotype distributions of these isolates makes it impossible to determine the representativeness of this isolate pool. If any additional data is available to determine how representative these isolates are of L. monocytogenes as a whole, they would be hugely helpful to this manuscript. If these isolates are all the same serotype or only represent a few over-represented clones of L. monocytogenes it will change the implications of the findings somewhat. The authors mention that efflux pumps could have been involved in the evolved phenotypes observed, Rakic-Martinez et al 2011 cited by the authors used the efflux pump inhibitor reserpine to test this same hypothesis in strains obtained under similar conditions, is this something the authors attempted or could attempt to better elucidate the nature of the changes they observed. This is a very subjective suggestion, but the inclusion of color into the figures in my opinion would improve the presentation but is by no means necessary. A few minor questions / possible corrections below:
-line 58 "biocides are indeed used to ensure microbiological quality requirements found on hygiene food law" is a bit awkward please revise for clarity
-line 61 "aiming at eliminate" aiming to eliminate? perhaps
-line 73 "various antibiotics as ciprofloxacin" such as ciprofloxacin
-line 263 "To resume," I think you might mean to conclude
Author Response
The authors survey a sizeable collection of L. monocytogenes from diverse sources and assayed their susceptibiltiy to antibiotics and biocides with the intent of identifying source specific trends in susceptibilities to these compounds of public health relevance. Overall the study is well conceceived and executed and well presented in their manuscript, I have only a few minor conceptual points and possible text corrections detailed below.
Thanks for your comments.
My first relates to the attributes of the strain pool, while they were assembled from different sources what is known of their genotypes as this can have significant impacts on functional attributes of L. monocytogenes. Not knowing even the serotype distributions of these isolates makes it impossible to determine the representativeness of this isolate pool. If any additional data is available to determine how representative these isolates are of L. monocytogenes as a whole, they would be hugely helpful to this manuscript. If these isolates are all the same serotype or only represent a few over-represented clones of L. monocytogenes it will change the implications of the findings somewhat.
We provided in supplementary material (table S3) the list of the 205 listeria strains we used with the year and source of isolation. As mentionned, strains were isolated from a wide variety of food products (meat, dairy products, ready-to-eat, fish, vegetables), various wild- or farm- animals and various environmental sources (soil, water, ...), over 16 years and from 13 european countries. These 205 strains were selected from the wide European collection of Listeria strains built in the framework of the “One Health European Joint Programme ListAdapt” (grant #773830; https://onehealthejp.eu/jrp-listadapt/) and extensively described in Félix, B.; Sevellec, Y.; Palma, F.; Felten, A.; Radomski, N.; Mallet, L.; Blanchard, Y.; Leroux, A.; Soumet, C.; Bridier, A.; et al. A European-Wide Dataset to Decipher Adaptation Mechanisms of Listeria Monocytogenes to Diverse Ecological Niches. Scientific Data 2020, in press. as mentioned in the article line 283. Such collection was constituted especially to illustrate the wide diversity of Lm strains present in environment.
The authors mention that efflux pumps could have been involved in the evolved phenotypes observed, Rakic-Martinez et al 2011 cited by the authors used the efflux pump inhibitor reserpine to test this same hypothesis in strains obtained under similar conditions, is this something the authors attempted or could attempt to better elucidate the nature of the changes they observed. This is a very subjective suggestion, but the inclusion of color into the figures in my opinion would improve the presentation but is by no means necessary.
The genetic and molecular basis of ciprofloxacin resistance in QAC-adapted strains will be addressed in a dedicated work and constitue the main perspective of this work. The sequencing of genomes for strains which became resistant to CIP and MIC assays in presence of efflux pumps inhibitors are actually planned to go further in the description of the mechanisms underlying the adaptation to QAC and CIP resistance development.
A few minor questions / possible corrections below:
-line 58 "biocides are indeed used to ensure microbiological quality requirements found on hygiene food law" is a bit awkward please revise for clarity
-line 61 "aiming at eliminate" aiming to eliminate? perhaps
-line 73 "various antibiotics as ciprofloxacin" such as ciprofloxacin
-line 263 "To resume," I think you might mean to conclude
All these points are modified accordingly in the revised manuscript.
Reviewer 4 Report
in this article, the authors address the important problem of the onset of resistance to biocides and antibiotics, referring in particular to Lm
in general, two major considerations:
- It would greatly help the reading if the acronyms (although universally adopted) were spelled out the first time they appear.
- however interesting and methodologically correct, the article is not very innovative and, as the authors themselves say in the conclusion, their results substantially confirm the results of other authors. I believe, however, that the work could be greatly enriched if what the authors themselves formulate as hypotheses to explain their results were verified (I refer to the increase in efflux pumps)
some minimal observations, in detail:
Line 112: How are “tentative ECOFF” established?
Line 116: you affirm that “that the origin of strains did not affect the antibiotic susceptibility profiles among the Lm stains panel (data not shown).” it seems to me too interesting a result not to be shown. Perhaps it deserves to be added at least in the supplementary
line 124: it might be useful to explain here what criterion was adopted to select the 28 strains on which to evaluate the effects of repeated exposure to biocides on MIC values
Figure 3 and 4 are hardly comprehensible, we suggest a violin plot representation which would help in the interpretation differences between the samples
Line 161, 197: please correct typo (stains)
Can the authors formulate a hypothesis about why the behavior between QAC and non-QAC is different, in terms of inducing resistance to CIP and other antibiotics?
Author Response
in this article, the authors address the important problem of the onset of resistance to biocides and antibiotics, referring in particular to Lm
in general, two major considerations:
- It would greatly help the reading if the acronyms (although universally adopted) were spelled out the first time they appear. This was modified in the revised manuscript.
- however interesting and methodologically correct, the article is not very innovative and, as the authors themselves say in the conclusion, their results substantially confirm the results of other authors. I believe, however, that the work could be greatly enriched if what the authors themselves formulate as hypotheses to explain their results were verified (I refer to the increase in efflux pumps).
The investigation of the genetic and molecular basis of ciprofloxacin resistance in QAC-adapted strains will be addressed in a dedicated work and constitue the main perspective of this work. However, we believe the present work contains original and interesting results to be published without such molecular analysis.
First, we used a large panel of strains (>200) to compare susceptibility of strains from different ecological niches (food, environment or animal) to a wide variety of antimicrobial compounds. In particular, we interestingly showed an impact of the strain origins on the MIC distribution for some biocides as QACs highlighting the potential adaptation of Listeria to such class of compounds in food area. It echoes the fact that an large proportion of strains we then adapted to QACs displayed a loss of susceptibility to ciprofloxacin suggesting a potential role of the adaptation to QACs on the food chain also in the resistance toward important antibiotics. Moreoever, for the first time we demonstrated a link between exposure to DDAC and ciprofloxacin resistance in Listeria.
some minimal observations, in detail:
Line 112: How are “tentative ECOFF” established?
Tentative ECOFF is defined by EUCAST as "ECOFFs (and TECOFFs) distinguish microorganisms without (wild type) and with phenotypically detectable acquired resistance mechanisms (non-wild type) to the agent in question. The epidemiological cut-off value is shown in the tables and the bottom left-hand corner of each MIC and zone diameter graph. TECOFFs (ECOFFs in parentheses) are based on 3 or 4 distributions and ECOFFs on at least 5 and up to 100 or more distributions."
Line 116: you affirm that “that the origin of strains did not affect the antibiotic susceptibility profiles among the Lm stains panel (data not shown).” it seems to me too interesting a result not to be shown. Perhaps it deserves to be added at least in the supplementary
As the MIC values distribution between the 200 strains is very narrow concerning the vast majority of antibiotics (only 3 dilutions for 8/11 antibiotics), such results are not included to not burden the article with non-informative data. Indeed, table 1 showed that the MIC values were highly homogeneous within the 205 listeria panel and we therefore did not found statistical difference between ecological origins of strains as in many case we found a similar MIC value for more than 80% of the strains.
line 124: it might be useful to explain here what criterion was adopted to select the 28 strains on which to evaluate the effects of repeated exposure to biocides on MIC values
A sentence was added line 127 in the revissed manuscript to clarify criterion of selection of the 28 strains.
Figure 3 and 4 are hardly comprehensible, we suggest a violin plot representation which would help in the interpretation differences between the samples
We used classic box and wiskers plots because that enables to represent quickly and visually the distribution trend, the median of distribution and to figure statistical differences between distributions. We are not convinced that violin plot will help here to better compare results as here MIC values are discrete variables.
Line 161, 197: please correct typo (stains)
This was corrected in revised manuscript.
Can the authors formulate a hypothesis about why the behavior between QAC and non-QAC is different, in terms of inducing resistance to CIP and other antibiotics?
The main hypothesis we formulated about the specific link between QAC and ciprofloxacin is mentionned in discussion from line 249 to 269. It relies on the overexpression of non-specific efflux pumps able to extrude both QACs and antibiotic such as ciprofloxacin as detailed in the manuscript.
Round 2
Reviewer 1 Report
If the measures of MIC conditions are changed from those for EUCAST, authors must be a need to show that there is no difference between gold standard methods (EUCAST and/or CLSI) and author’s modified methods. (in section of method and results)
Authors did not show the enough dates and information.
Author Response
If the measures of MIC conditions are changed from those for EUCAST, authors must be a need to show that there is no difference between gold standard methods (EUCAST and/or CLSI) and author’s modified methods. (in section of method and results)
Authors did not show the enough dates and information.
As already mentionned in our previous reply to reviewer's comments, the aim of the present study was to compare a large panel of Listeria monocytogenes strains for their antibiotic and biocide profiles and the potential role of biocides in antibiotic resistance emergence. In this aim, we adapted slightly the EUCAST protocol because we designed specific custom plates for Listeria. MIC determination was performed by microdilution broth in the Sensititre TREK system and to facilitated reading of MIC results, we extended incubation from 20h to 24h.
Once again, we assumed such modifications did not alter extensively MIC values based on our experience with Listeria in these conditions, that only facilitated the reading of the plate since we processed a large number of strains with various behaviours. Anyway, we did not state in the manuscript that we performed MIC determination according formally to EUCAST protocol. Indeed, the importants results of the study was the evolution of CIP MIC after QAC exposure. Antibiotic MIC were determined to obtain a basis level to evaluate MIC modification after biocide exposure, and were determined strictly in identical conditions.
Nevertheless, we modified the manuscript from line 104 to line 120 and also in Material/method section line 302-303 to underline that we did not formally use the EUCAST protocol for MIC determination as we extended incubation from 20h to 24h.